## Research Article

**Key words:**
α-Synuclein; fission; lipid membrane; membrane remodelling

**Author for correspondence:**
*Katarzyna Makasewicz,
E-mail: k.makasewicz@gmail.com;
Emma Sparr,
E-mail: emma.sparr@fkem1.lu.se

# α-Synuclein-induced deformation of small unilamellar vesicles

Katarzyna Makasewicz[1]* Stefan Wennmalm[2], Sara Linse[3] and Emma Sparr[1]*

[1]Department of Physical Chemistry, Lund University, 221 00 Lund, Sweden; [2]Department of Applied Physics, SciLifeLab, Royal Institute of Technology, 171 65 Solna, Sweden and [3]Department of Biochemistry and Structural Biology, Lund University, 221 00 Lund, Sweden

## Abstract

α-Synuclein is a small neuronal protein that reversibly associates with lipid membranes. The membrane interactions are believed to be central to the healthy function of this protein involved in synaptic plasticity and neurotransmitter release. α-Synuclein has been speculated to induce vesicle fusion as well as fission, processes which are analogous to each other but proceed in different directions and involve different driving forces. In the current work, we analyse α-synuclein-induced small unilamellar vesicle deformation from a thermodynamics point of view. We show that the structures interpreted in the literature as fusion intermediates are in fact a stable deformed state and neither fusion nor vesicle clustering occurs. We speculate on the driving force for the observed deformation and put forward a hypothesis that α-synuclein self-assembly on the lipid membrane precedes and induces membrane remodelling.

## Introduction

Lipid bilayers are a foundation of a living cell and its organelles, including nucleus, endoplasmic reticulum, Golgi apparatus and mitochondria. The bilayers separate different compartments from each other and allow for selective transport of signalling substances. The specific geometry, topology and dynamics of a membrane are strongly connected to its biological functions. Membrane remodelling events are thus a vital part of the processes through which the lipid bilayer carries out its functions. Some membrane remodelling processes, such as deformation into non-spherical objects, are not expected to involve high energy intermediates, while others such as fusion or fission generally involve a higher energy barrier (Seifert, 1997; Kozlov *et al.*, 2010). The free energy of the system is affected by the properties of the membrane, which depend on its composition with respect to both lipids and proteins. Lipid species of specific molecular geometry may favour non-planar structures and as a result facilitate membrane deformation. Various cytosolic proteins associate with the membrane, change its properties and induce membrane remodelling (Farsad and De Camilli, 2003; Prinz and Hinshaw, 2009; Suetsugu *et al.*, 2014).

α-Synuclein is a 140 amino acid presynaptic protein thought to be implicated in neurotransmitter release – a process which involves large-scale membrane remodelling. The N-terminal domain of α-synuclein mediates membrane binding of the protein. On a membrane, parts or all of the N-terminal domain fold into an amphipathic α-helix (Bodner *et al.*, 2009). The hydrophilic and hydrophobic residues are found on the opposite faces of the helix, flanked by charged residues (Davidson *et al.*, 1998). The amphipathic helix is a common feature of membrane-binding and membrane-deforming proteins' structure (Zhukovsky *et al.*, 2019).

There are several reports suggesting that α-synuclein induces fusion of small unilamellar vesicles (SUVs) (Georgieva *et al.*, 2010; Fusco *et al.*, 2016), while other studies describe that the protein stabilises vesicle curvature thereby preventing fusion (Kamp and Beyer, 2006; Kamp *et al.*, 2010; Braun *et al.*, 2012). These different roles proposed for α-synuclein involve exactly opposite driving forces. Thus, the mechanism of action of α-synuclein and the direction of the induced membrane deformation has yet to be clarified.

In this work, we studied α-synuclein-induced deformation of SUVs (diameter ca. 150 nm) composed of 1,2-dioleoyl-sn-glycero-3-phosphocholine:1,2-dioleoyl-sn-glycero-3-phospho-L-serine 7:3 (DOPC:DOPS 7:3 molar ratio) from a thermodynamics point of view. We analysed *cryo*-TEM images of deformed vesicles at different time points after mixing protein and vesicles, for systems with different lipid compositions and different lipid-to-protein (L/P) ratios. In order to get quantitative information on the size and number of the vesicles, *cryo*-TEM studies were combined with fluorescence cross-correlation spectroscopy (FCCS), inverse fluorescence cross-correlation spectroscopy (iFCCS) and nanoparticle tracking analysis (NTA). Based on these experiments, we can rule out that vesicle fusion or vesicle clustering occurs upon α-synuclein adsorption, while fission may occur at some of the conditions investigated. We first discuss our findings in terms of

the thermodynamic and kinetic stability of the deformed vesicles. We then speculate on the driving forces for deformation as well as the possible molecular mechanism behind the $\alpha$-synuclein-induced membrane remodelling.

## Results

### $\alpha$-Synuclein induces deformation of SUVs

The majority of DOPC:DOPS 7:3 SUVs are spherical in shape in our preparation, as illustrated in the *cryo*-TEM images (Fig. 1*a*). Upon the addition of $\alpha$-synuclein, the vesicles undergo deformation, changing shape to elongated spheroid structures, in some cases with appearance of equal-sized pearls-on-a-necklace (Fig. 1*b*). The vesicles appear deformed already at 25 s after the addition of $\alpha$-synuclein (Supplementary Fig. 1), which is the dead-time from sample mixing through deposition on a carbon grid to plunging in liquid ethane. The vesicles remain deformed for an extended period of time, at least for 24 h, after the addition of the protein (Fig. 1*b*). The same trend is seen when $\alpha$-synuclein is added to vesicles containing only one lipid component, DOPS, thus ruling out that the observed deformation is due to lipid segregation within the vesicle (Fig. 1*d*). In all of the protein-containing samples, $\alpha$-synuclein was added to attain a lipid-to-protein (L/P) ratio of 200. We have previously shown that for the DOPC:DOPS 7:3

system at this L/P virtually all $\alpha$-synuclein molecules are bound to the membrane (Makasewicz *et al.*, 2021).

We note that there are a few examples of deformed vesicles present in the sample containing only vesicles in MES buffer (Fig. 1*a,c*), which can be explained by that some amount of water evaporates from the thin liquid film during *cryo*-TEM sample deposition. The evaporation leads to a hypertonic osmotic imbalance across the lipid membrane that may cause vesicle deformation (Dubochet *et al.*, 1988; Pencer *et al.*, 2001; Zong *et al.*, 2018). We therefore prepared DOPC:DOPS 7:3 SUVs in pure water and for this case we observed only spherical vesicles in *cryo*-TEM images (Fig. 1*e*). The SUVs prepared in water also undergo deformation to spheroid structures upon the addition of $\alpha$-synuclein and remain deformed for at least 24 h (Fig. 1*f*).

### Fusion, fission or deformation?

The first question to be addressed in order to characterise $\alpha$-synuclein-induced vesicle deformation is whether the structures observed with *cryo*-TEM are fusion intermediates, fission intermediates or stable non-spherical structures. To address this question, we designed a fluorescence cross-correlation spectroscopy (FCCS) experiment. In FCCS, the amplitude of the red auto-correlation curve contains information on the number of red particles in the detection volume and the amplitude of the green auto-correlation curve contains information on the number of green particles in the detection volume. The amplitude of the cross-correlation curve together with that of the two auto-correlation curves contains information on the number of particles that are both green and red.

We first carried out the FCCS experiment on the 1:1 mixture of SUVs containing a green-fluorescent lipid analogue and SUVs containing a red fluorescent lipid analogue in the absence of $\alpha$-synuclein. Fig. 2*a* (top left) shows representative auto-and cross-correlation curves. The amplitude of the green auto-correlation curve is lower than the amplitude of the red auto-correlation curve due to higher "background" fluorescence from the residual green dye. After the correction for the contribution from the residual free dye (for details see Materials and Methods), the number of green particles and the number of red particles are equal within the experimental error as shown in Fig. 2*c*. This implies that the numbers of green and red vesicles in the sample are equal (Fig. 2*c*). The amplitude of the cross-correlation curve is close to zero, which implies that there are no vesicles in the sample that are both red and green. Next, we added non-fluorescently labelled $\alpha$-synuclein to the sample composed of the 1:1 mixture of red and green vesicles. If fission occurred upon the addition of $\alpha$-synuclein, the number of red- and green-labelled particles in the sample would increase. If fusion occurred, the doubly-labelled particles are expected to emerge. The experiment was performed for samples at L/P 50, 100 and 200, and the data are shown in Fig. 2*a*. For all samples, it is clear that the amplitude of the cross-correlation function is unchanged with respect to the protein-free sample. This implies that there are no objects that are both green and red. The numbers of green and red vesicles, $N_g$ and $N_r$, extracted from the background corrected amplitudes of the auto-correlation curves are approximately the same in all samples, as shown in Fig. 2*c*. This implies that no detectable fusion, fission nor vesicle clustering is induced by $\alpha$-synuclein.

In order to corroborate the findings from FCCS, we employed nanoparticle tracking analysis (NTA). NTA is a technique allowing to estimate the number of particles in the sample, which does not rely on fluorescent labelling. We measured the number of particles in the

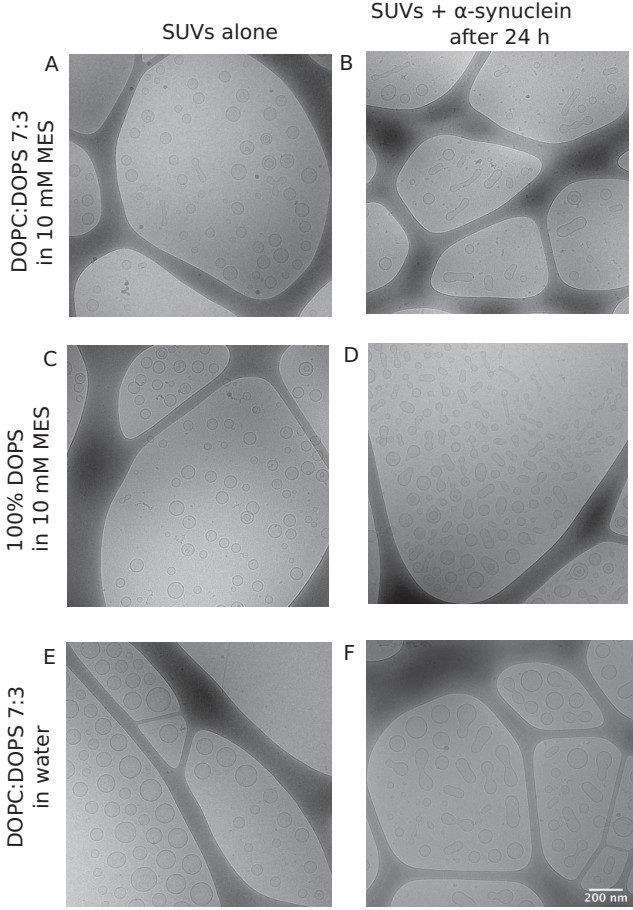

**Figure 1.** $\alpha$-Synuclein induces deformation of SUVs. *Cryo*-TEM images of SUVs without and with $\alpha$-synuclein at L/P 200. (*a,b*) DOPC:DOPS 7:3 SUVs in 10 mM MES buffer pH 5.5. (*c,b*) 100% DOPS SUVs in 10 mM MES buffer pH 5.5. (*e,f*) DOPC:DOPS 7:3 SUVs in pure water. The scale bar is shown in panel *f* and is the same for all images.

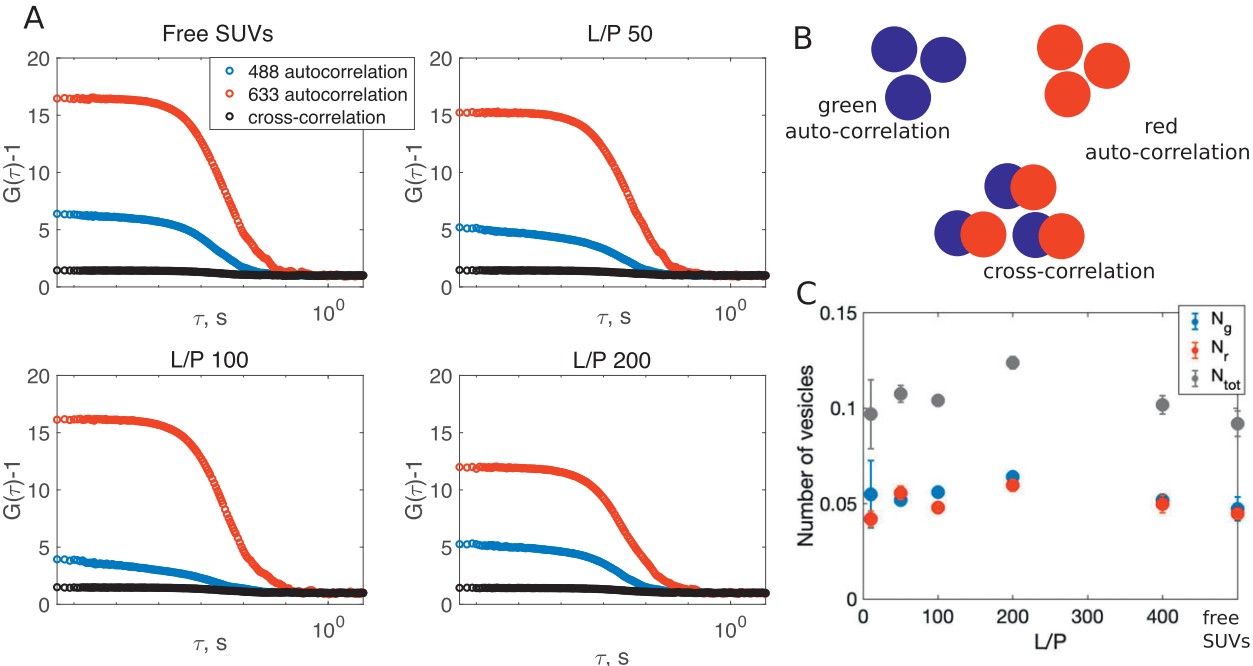

**Figure 2.** Detecting fusion, fission or vesicle clustering with FCCS. (*a*) The auto- (green and red) and cross-correlation (black) curves for samples composed of a 1:1 mixture of green and red SUVs with no added protein, and for samples containing SUVs with α-synuclein at L/P 50, 100 and 200. An almost zero amplitude of the cross-correlation curve indicates that none of the samples contains doubly labelled species. (*b*) Cartoons representing the fluorescent species whose presence is reflected in the auto- and cross-correlation curves. The number of green/red species in inversely proportional to the green/red auto-correlation amplitude, while the number of green–red species is directly proportional to the cross-correlation amplitude. (*c*) Average number of green and red vesicles $\pm$ SD in the focal volume extracted from the green and red auto-correlation amplitudes corrected for the contribution from the residual free dye (higher amount of residual free dye for green-fluorescent lipids than for red fluorescent lipids). The total number of vesicles $\pm$ SD was calculated as the sum of $N_g$ and $N_r$. The fact that the average number of vesicles in the focal volume is lower than 1 means that SUVs are present in the focus for only a fraction of the measurement time.

sample for protein-free DOPC:DOPS 7:3 SUVs and for SUVs with α-synuclein at L/P 25, 50, 100 and 200. The number of vesicles in the samples containing vesicles with and without α-synuclein is equal within the experimental error (Supplementary Table 1). Thus, NTA results corroborate the finding that neither fission nor fusion of the vesicles occurs upon the addition of α-synuclein, which would result in a change of the number of vesicles in the sample.

### Change in vesicle volume upon the addition of α-synuclein

We employed iFCCS to determine the vesicle volume change upon the addition of α-synuclein. The model system for this experiment consisted of SUVs containing a green fluorescent lipid analogue (0.5% Oregon Green 488 DHPE), unlabelled α-synuclein and red fluorescent dye (Alexa Fluor 647) dissolved in the buffer to a concentration of 100 μM. A vesicle passing through the observation volume gives rise to a positive signal in the green intensity trace (Fig. 3*a*, bottom). Since the dye does not penetrate the lipid membrane, a diffusing vesicle displaces a fraction of dye molecules from the detection volume, which gives rise to a negative peak in the red intensity trace (Fig. 3*a*, top). The cross-correlation of the fluctuations of the positive intensity signal in the green channel with the fluctuations of the negative signal in the red channel gives rise to an anti-correlation curve (Fig. 3*b*; Wennmalm *et al.*, 2009; Wennmalm and Widengren, 2010). The amplitude of the anti-correlation curve is proportional to the volume of the particle (Wennmalm and Widengren, 2010). We have carried out the iFCCS experiments for free SUVs and SUVs with α-synuclein at L/P 25, 50, 100 and 200. The amplitude of the iFCCS curve recorded for the samples containing α-synuclein amounts to approx. 70% of the iFCCS amplitude of the

sample containing only SUVs. This implies that the volume of the vesicles decreases upon the addition of the protein to approx. 70% of the original volume (Fig. 3*c*). This is consistent with the type of deformation observed with *cryo*-TEM occurring under constant membrane surface area, although direct quantification of volumes cannot be done from the 2D microscopy images. An increased diffusion time of the vesicles with α-synuclein (Fig. 3*d*) is ascribed to the added mass and volume of the associated protein. Due to the very low permeability of the lipid membrane to the charged buffer molecules (Evans and Wennerström, 1999), the buffer concentration increases inside the vesicles when their internal volume decreases upon deformation. The osmotic imbalance induced by the shape change would be released if the vesicle went back to its original spherical shape. Thus, the deformation induced by α-synuclein binding must be highly favourable as the osmotic effect due to increase in buffer concentration inside the vesicle opposes the deformation and drives the relaxation to a spherical shape.

### The extent of vesicle deformation as a function of L/P

We used *cryo*-TEM to investigate whether there are any differences in the morphology of the deformed vesicles at different L/Ps. We imaged samples consisting of DOPC:DOPS 7:3 SUVs with α-synuclein over L/P range 25–200. Indeed, in the samples at L/P 25 and 50, we observed very long unduloid structures with an aspect ratio above 30, which are absent in the samples at L/P 100. In order to quantify the differences between the samples, we measured the aspect ratio for approx. 100 vesicles for each L/P ratio. The results of this analysis are presented in the box-whiskers plot in Fig. 4*a*.

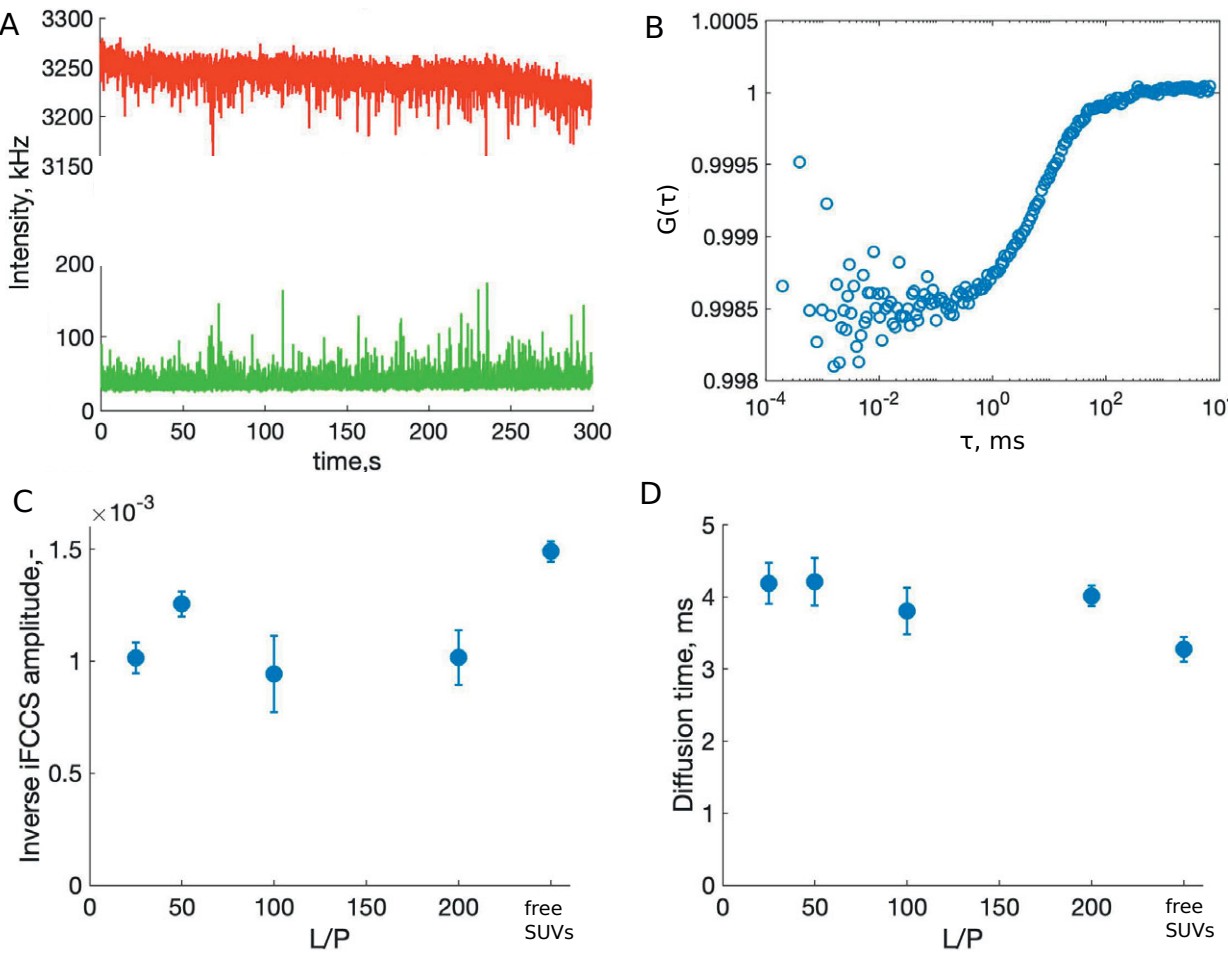

**Figure 3.** Determining the change in volume of the vesicles upon α-synuclein binding with iFCCS. (*a*) Green-labelled vesicles (0.5% Oregon Green 488 DHPE) give rise to a positive signal in the green intensity trace (bottom) and to a negative signal in the red intensity trace (top) due to displacement of red fluorescent dye from the detection volume. (*b*) An example of an anti-correlation curve. The fluctuations of the signals from the green and red channels are cross-correlated giving rise to an anti-correlation function whose amplitude is proportional to the volume of the vesicles. The iFCCS curves for all samples are shown in Supplementary Fig. 2. (*c*) The inverse of the amplitude of the iFCCS function ± SD as a function of L/P ratio. (*d*) Diffusion time of the vesicle ± SD as a function of L/P ratio extracted from the auto-correlation curves in the green channel.

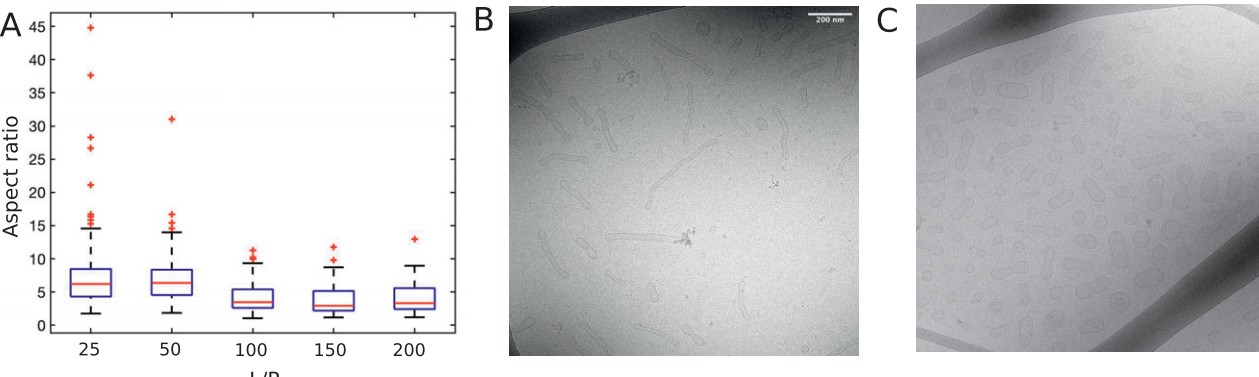

**Figure 4.** The extent of SUV deformation as a function of the lipid-to-protein ratio (L/P). (*a*) Box-whiskers plot of the aspect ratio of DOPC:DOPS 7:3 SUVs with α-synuclein imaged with *cryo*-TEM as a function L/P. The aspect ratio was calculated as the ratio between the long and the short axis of the vesicle. For each L/P, 100 vesicles were analysed. (*b*) *Cryo*-TEM image of DOPC:DOPS 7:3 SUVs with α-synuclein at L/P 50. (*c*) *Cryo*-TEM image of DOPC:DOPS 7:3 SUVs with α-synuclein at L/P 200. The scale bar for both images is the same and is shown in panel *b.*

### Effect of lipid composition

Using FCCS, NTA and iFCCS, we have established that neither fusion nor vesicle clustering occurs upon α-synuclein addition to DOPC:DOPS 7:3 SUVs in MES buffer. We have also carried out the analysis of the 2D cross-section areas of the SUVs in the presence and absence of α-synuclein. The results are plotted in the histogram in Fig. 5*a*. The cross-section areas of the vesicles after the addition of the protein did not increase, which is consistent with no fusion taking place in the system. The emergence of a small population of

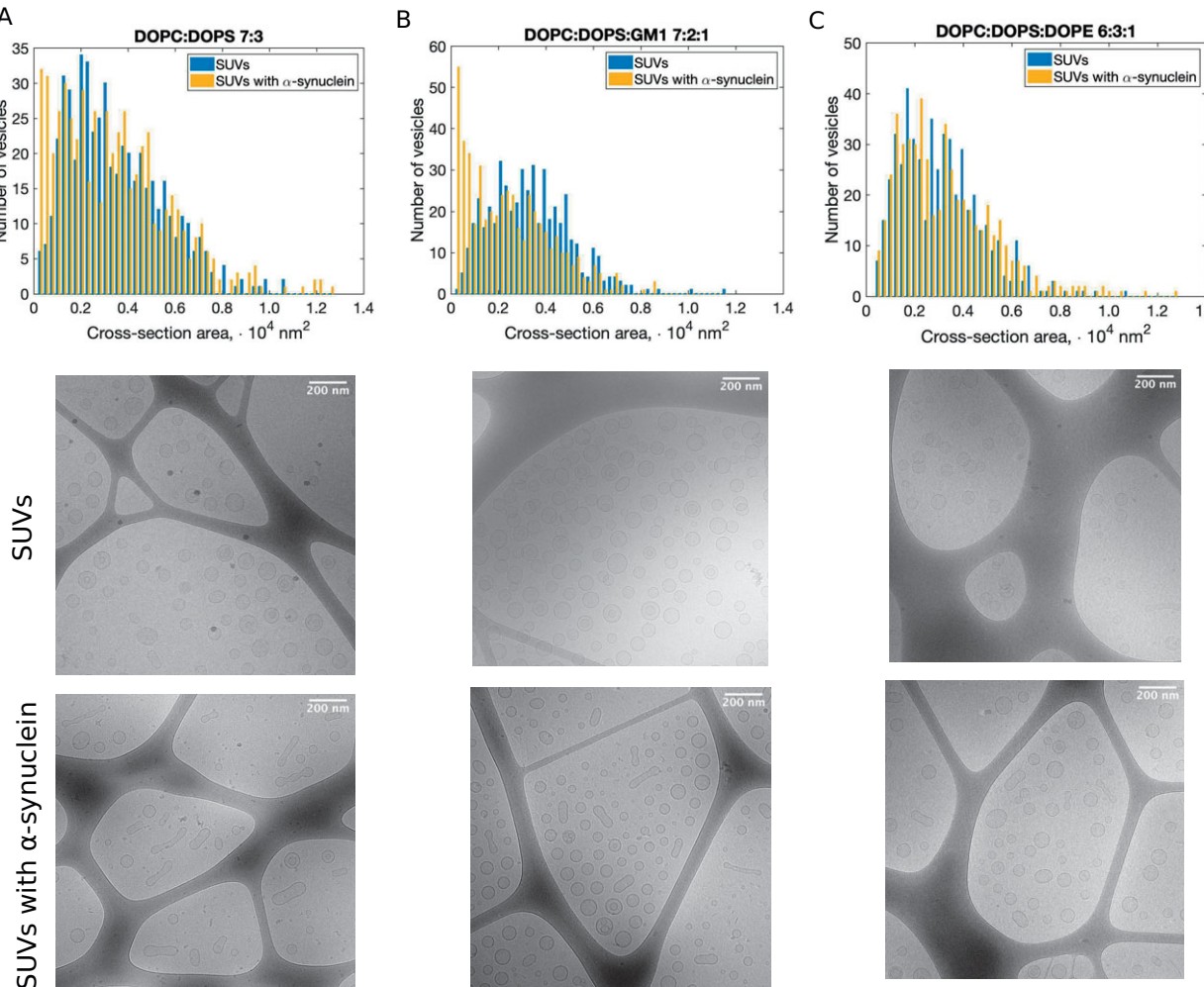

**Figure 5.** Cross-section area analysis of SUVs without and with α-synuclein imaged with *cryo*-TEM. (*a,b,c*) Histogram of cross-section areas of SUVs composed of DOPC:DOPS 7:3, DOPC:DOPS:GM1 7:2:1 and DOPC:DOPS:DOPE 6:3:1, respectively. Examples of the analysed images are shown below the histograms. The samples containing SUVs and α-synuclein (L/P 200) were imaged 24 h after the addition of the protein. For each sample, 550 vesicles from different images were analysed.

vesicles with lower cross-section areas suggests that fission have occurred for a small number of vesicles, which could not be detected in the FCCS and NTA experiments.

In order to gain more insight into the role of membrane composition in the deformation process, we incorporated lipid molecules that favour non-planar structures into the DOPC:DOPS vesicles. We chose two lipid compositions, DOPC:DOPS:GM1 7:2:1 and DOPC:DOPS:DOPE 6:3:1, with the same overall charge as the DOPC:DOPS 7:3 mixture. The spontaneous curvature of the bilayers formed from all mixtures equals one, assuming homogeneous distribution of all lipid species in the inner and outer leaflet. However, the modulus of Gaussian curvature, $\bar{\kappa}$, is expected to change in the opposite direction upon incorporation of a fraction of GM1 or a fraction of DOPE in the DOPC:DOPS mixture, due to the different molecular shapes of those lipids. $\bar{\kappa}$ affects the energy of fusion/fission intermediates and thus the height of the energetic barriers that have to be crossed (Kozlov *et al.*, 2010).

We have imaged SUVs of these compositions with *cryo*-TEM in the absence and presence of α-synuclein. The results of the cross-section area analysis are shown in Fig. 5b,c. The distribution of the cross-section areas of the vesicles composed of DOPC:DOPS:GM1 7:2:1 after the addition of α-synuclein is skewed towards values of low cross-section areas as compared to the protein-free system.

This indicates that fission may be favoured for lipid systems containing a fraction of GM1. The comparison of the cross-section areas of the DOPC:DOPS:DOPE 6:3:1 SUVs reveals no significant changes between the bare vesicles and the vesicles with α-synuclein (Fig. 5c), thus indicating that neither fusion nor fission has occurred in this system.

We note that the inherent complication in the analysis of the vesicle cross-section areas from the *cryo*-TEM images is that in a 2D image we are not able to distinguish between spheres and prolates imaged from the short end. Nevertheless, the emergence of a population of small objects for certain lipid compositions (DOPC:DOPS 7:3, DOPC:DOPS:GM1 7:2:1) after the addition of the protein and not for others (DOPC:DOPS:DOPE 6:2:1) implies that fission is facilitated in the former systems.

## Discussion

The results of the current study show that α-synuclein adsorption to spherical SUVs induces deformation to a spheroid shape and the vesicles remain in the deformed state for an extended period of time (Fig. 1). Such deformed vesicles have previously been interpreted as fusion intermediates (Fusco *et al.*, 2016; Man *et al.*, 2020), but may

as well be fission intermediates or stable deformed structures. We have carried FCCS, iFCCS and NTA experiments to identify the nature of the deformed vesicular structures observed with *cryo*-TEM. Our results imply that neither fusion nor fission occurs in the DOPC:DOPS 7:3 system upon the addition of $\alpha$-synuclein to a detectable extent. The iFCCS results show that the volume of the vesicles after the addition of $\alpha$-synuclein is lower than that of the free vesicles (Fig. 3*c*). The decrease in volume is due to the deformation from sphere to spheroid with the constraint of constant membrane surface area.

An important question is why do the vesicle deformation persists for an extended period of time. The *cryo*-TEM samples of vesicles with $\alpha$-synuclein were prepared at 25 s (Supplementary Fig. 1) and 24 *h* after the addition of the protein (Fig. 1). The fact that the deformation persists for an extended period of time implies that the deformed vesicle is a kinetically stable state, although it might not be thermodynamically stable.

The deformation of vesicles upon $\alpha$-synuclein binding is expected to be related to the fact that the adsorption of protein leads to an asymmetry in the originally symmetric lipid bilayer. Such asymmetry generates spontaneous curvature that deviates from the curvature of the bare lipid membrane. The curvature generated by $\alpha$-synuclein binding is positive (towards the surrounding medium)(Zemel *et al.*, 2008; Varkey *et al.*, 2010; Braun *et al.*, 2012; Lipowsky, 2013; Westphal and Chandra, 2013). At some critical bilayer coverage, the spontaneous curvature of the outer monolayer will become higher than the curvature of the original vesicle. One way for a vesicle to relax this curvature energy is to deform and/or undergo fission, which gives two vesicles of higher curvature than the original vesicle. However, fission involves a disruption in membrane continuity and thus a large energy barrier (Kozlov *et al.*, 2010), which may not be overcome by thermal fluctuations under the conditions of our experiments. A consequence of the presence of the high-energy barrier may be the formation of kinetically stable structures reminiscent of fission precursors, such as the deformed vesicles observed with *cryo*-TEM. The same argument can be used to show that fusion is not thermodynamically possible in the studied system. For the fusion to be favoured, $\alpha$-synuclein adsorption would need to generate a decrease in the spontaneous curvature. However, the existence of a driving force for fusion is not consistent with the formation of stable deformed vesicles. No high free-energy barrier is expected between a fusion intermediate and a spherical vesicle. Thus, there is no reason why already fused vesicles would remain deformed instead of relaxing to a spherical shape.

Altogether, our data suggest that the spheroid-shaped vesicles observed with *cryo*-TEM are fission intermediates and that fusion, a process occurring in the opposite direction, is not favoured in the studied system.

### The role of the density of bound $\alpha$-synuclein on the membrane

It is expected that the extent of vesicle deformation increases with the density of the bound protein on the membrane (Lipowsky, 2013). High density of bound protein provides additional bending energy, which leads to higher curvature of the resulting structures. Our analysis of the *cryo*-TEM images of samples at different L/P ratios revealed that indeed the aspect ratio of the vesicles in the presence of $\alpha$-synuclein increases with decreasing L/P (Fig. 4*a*). The lower the L/P, the higher the curvature of the deformed vesicles. This result is consistent with the observation of Varkey *et al.* (2010) that the extent of tubulation of vesicles composed of another

anionic phospholipid, 1-palmitoyl-2-oleoyl-sn- glycero-3-[phospho-RAC-(1-glycerol)] (POPG), in the presence of $\alpha$-synuclein depends on L/P.

The high curvature of the membrane induced by densely bound $\alpha$-synuclein molecules may be important for the process of vesicle budding. We note, however, that *in vivo*, vesicle fission from the cell membrane is orchestrated by a large amount of proteins (Yang *et al.*, 2006; Morlot and Roux, 2013), which take part in different stages of the process and thus it is possible that $\alpha$-synuclein deforms the membrane but does not cause fission on its own.

### Protein organisation on the membrane

The binding energy between a protein and a lipid membrane is much lower than the energy required for large-scale changes in membrane geometry (Reynwar *et al.*, 2007; Corey *et al.*, 2020). This implies that a cooperative action of many proteins is necessary for membrane remodelling on a mesoscale. The organisation of proteins on the membrane leads to the formation of a supramolecular structure whose net effect on the membrane curvature is different than the sum of its parts. Protein self-assembly on a membrane has been widely studied in the context of e.g. clathrin-mediated endocytosis where multiple proteins form so-called "coats" on the membrane and cause it to bend (Kaksonen and Roux, 2018), but is not well understood in the case of $\alpha$-synuclein. Computational and experimental studies on $\alpha$-synuclein tubulation of vesicles revealed that a very important aspect of membrane remodelling by this protein is its uniquely long amphipathic $\alpha$-helix in the membrane-bound state (Braun *et al.*, 2012, 2017). Based on coarse-grained MD simulations, Braun *et al.* proposed that local perturbations in the membrane induced by bound $\alpha$-synuclein molecules reinforce each other, which leads to large-scale membrane remodelling. Other experimental and computational studies on membrane remodelling proteins suggest that protein organisation on a membrane precedes membrane remodelling (Simunovic *et al.*, 2013, 2018). This has also been hypothesised to be the case for $\alpha$-synuclein but has yet to be proved or disproved experimentally. Here, we show that deformation inevitably accompanies $\alpha$-synuclein adsorption to the outer leaflet of SUVs. Since $\alpha$-synuclein binding to lipid membranes has been found to be highly positively cooperative (Makasewicz *et al.*, 2021), the hypothesis that the protein self-assembles on the membranes seems reasonable. The protein self-assembly is expected to involve attractive protein–protein interactions, while additional contributions may arise from the perturbations that a bound protein induces in the properties of the membrane.

### Conclusions

We have analysed $\alpha$-synuclein-induced deformation of SUVs in terms of the kinetic and thermodynamic stability of the deformed structures. Based on our experimental data, we conclude that neither fusion nor clustering of the vesicles occurs upon $\alpha$-synuclein addition to SUVs of the compositions investigated here. The deformed vesicles observed with *cryo*-TEM are kinetically stable fission intermediates consistent with fission involving an energy barrier that may not be overcome by thermal fluctuations under the conditions investigated here. We show however that fission of the vesicles upon $\alpha$-synuclein adsorption can be facilitated by the incorporation of ganglioside lipids in the membrane.

The presented results contribute to the understanding of the healthy function of α-synuclein, which involves synaptic vesicle trafficking. We put forward a hypothesis that α-synuclein self-assembles on a membrane and forms a supra-molecular structure with a higher spontaneous curvature than the curvature of the membrane. This causes the membrane to bend, which is a necessary step in vesicle budding.

## Methods

### α-Synuclein expression and purification

α-Synuclein of human wild-type sequence was expressed in *Escherichia coli* from synthetic genes with *E. coli*-optimised codons cloned in the Pet3a plasmid (purchased from Genscript, Piscataway, NJ). The protein was purified using heat treatment, ion-exchange and gel filtration chromatography, as previously described (Grey *et al.*, 2011). The purified protein was stored as multiple identical aliquots at −20℃. All experiments started with gel filtration of such aliquots on a 10 × 300 mm Superdex 75 column (GE Healthcare) to isolate fresh monomer in 10 mM MES buffer at pH 5.5. For the *cryo*-TEM experiment in pure water α-synuclein was desalted using a 5 mL HiTrap desalting column (GE Healthcare).

### Lipids

Liophilized lipids: 1,2-dioleoyl-sn-glycero-3-phospho-L-serine sodium salt (DOPS), 1,2-dioleoyl-sn-glycero-3-phosphocholine (DOPC), 1,2-dioleoyl-sn-glycero-3-phosphoethanolamine (DOPE), Oregon Green 488 1,2-dihexadecanoyl-sn-glycero-3-phosphoethanolamine (DHPE-488) and Ganglioside GM1 from ovine brain were purchased from Avanti Polar Lipids (Alabaster AL). Liophilized Atto-647 1,2-dipalmitoyl-sn-glycero-3-phosphoethanolamine (DPPE-647) was purchased from ATTO-TEC GmbH.

### SUV preparation

SUVs were prepared by extrusion using Avanti Mini Extruder (Avanti Polar Lipids). The desired volume of the lipid mixture in chloroform:methanol (7:3 volume ratio) was left overnight in a vacuum oven at room temperature for the solvent to evaporate. The dried lipids were then hydrated with 10 mM MES buffer at pH 5.5 or with pure MilliQ water and left on stirring for 2 h at room temperature. The SUVs were obtained by extruding 21 times through 100 nm pore size filters (200 nm pore size filters for iFCCS experiment) that had been saturated with the same lipids before use.

### Cryo-TEM

*Cryo*-TEM samples were prepared as follows. Lyophilized and desalted α-synuclein was resuspended in SUV dispersion. The lipid concentration was 20 mM in samples at L/P 200, 150, 100, 50, and 15 mM in the sample at L/P 25. The protein concentration was varied accordingly to reach the desired L/P ratio.

Specimens for *cryo*-TEM were prepared in an automatic plunge freezer system (Leica EM GP). The climate chamber temperature was kept at 21℃, and relative humidity was 90% to minimise loss of solution during sample preparation. The specimens were prepared by placing 4 μl solution on glow discharged lacey formvar carbon-coated copper grids (Ted Pella) and blotted with filter paper before being plunged into liquid ethane at −180℃. This leads to vitrified specimens, avoiding component segmentation and rearrangement, and the formation of water crystals, thereby preserving original microstructures. The vitrified specimens were stored under liquid nitrogen until measured. A Fischione Model 2550 cryo transfer tomography holder was used to transfer the specimen into the electron microscope, JEM 2200FS, equipped with an in-column energy filter (Omega filter), which allows zero-loss imaging. The acceleration voltage was 200 kV and zero-loss images were recorded digitally with a TVIPS F416 camera using SerialEM under low dose conditions with a 10 eV energy selecting slit in place.

### Cryo-TEM image analysis

The cross-section areas of the SUVs in the *cryo*-TEM images (Fig. 5) were analysed using ImageJ. For each sample, the cross-section areas of 550 SUVs from different images from the same experiment were measured. The aspect ratios (Fig. 4) were calculated form the measured distances of the long and short axes of the vesicles. For each sample 100 SUVs from different images from the same experiment were analysed.

### Fluorescence correlation spectroscopy (FCS)

FCS measurements were performed on Zeiss 780 and Zeiss 980 confocal laser scanning microscopes equipped for FCS, with a Zeiss water immersion objective, C-Apochromat 40×/1.2 NA. Samples were excited at 488 nm and 633 nm (Zeiss 780) or 639 nm (Zeiss 980) and fluorescence emission was collected at 499–622 nm and 641–695 nm, respectively. On the Zeiss 780, HiLyte 647 (D = 320 μm²/s; Wennmalm *et al.*, 2015) yielded $\tau_D = 64$ μs, $\omega = 0.31$ μm and $V = 0.96$ fL. 488 nm excitation of Alexa 488 (D = 414 μm²/s; Petrášek and Schwille, 2008) with detection of emission between 499–620 nm yielded $\tau_D = 32$ μs, $\omega = 0.23$ μm and $V = 0.36$ fL. On the Zeiss 980 LSM, measurement on Alexa 488 in solution (D = 414 μm²/s; Petrášek and Schwille, 2008) yielded $\tau_D = 21$ μs, $\omega = 0.186$ μm and $V = 0.177$ fL, and measurement on Alexa 633 (D = 340 ²/s) using 639 nm excitation yielded $\tau_D = 37$ μs, $\omega = 0.221$ μm and $V = 0.30$ fL. On the Zeiss 980 a pinhole of 15 μm diameter was used to minimise the detection volume of the 639 focus and thereby optimise the sensitivity of iFCCS.

### FCCS

The experimental system consisted of DOPC:DOPS 7:3 SUVs with 0.5% Oregon Green 488 DHPE (DHPE-488) (green-labelled vesicles) and DOPC:DOPS 7:3 SUVs with 0.5% Atto-647 DPPE (red-labelled vesicles). The green and red-labelled vesicles were mixed 1:1 to reach a total concentration of 40 μM. FCCS experiments were carried out on free vesicles and for vesicles with non-fluorescently labelled α-synuclein at L/P 50, 100, 200 and 400. The amplitudes of the green and red auto-correlation curves are not equal due to higher amount of residual free dye in the green channel. Background correction of the green ($A_g$) and the red ($A_r$) auto-correlation amplitudes was done as follows. From each measurement the intensity histogram of the total detected signal ($I_t$) was analysed, where the centre position of the main peak was taken as the mean background signal ($I_b$). $A_g$ and $A_r$ were then corrected by multiplication by $(I_{t,a})^2/(I_{t,a} - I_{b,a})^2$, where index a indicates g (green) or r (red). The number of the particles in the focal volume is calculated as the inverse of the background-corrected amplitude of the auto-correlation function.

## iFCCS

The model system consisted of 450 μM DOPC:DOPS 7:3 SUVs with 0.5% Oregon Green 488 DHPE (DHPE-488) (green-labelled vesicles), unlabelled α-synuclein at concentrations in the range 2–18 μM and 100 μM Alexa Fluor 647 NHS ester dye dissolved in the buffer. The vesicle diffusing through the focal volume displaces fluorescent dye molecules. The fluorescence intensity fluctuations due to the dye and due to the vesicles are correlated and the amplitude of the cross-correlation curve is proportional to the volume of the vesicle. The derivation of the relationship between the amplitude of the iFCCS curve and the particle volume is presented in SI.

## Nanoparticle tracking analysis

Free DOPC:DOPS 7:3 SUVs and SUVs with α-synuclein at L/P 25, 50, 100 and 200 samples were analysed using nanoparticle tracking analysis. In all samples, the lipid concentration was 500 nM, and the protein concentration was varied accordingly. The samples containing α-synuclein were measured 15 min after the protein addition. The measurements were carried out with a Nanosight LM10 instrument (Nanosight) equipped with a 405 nm laser. For each sample, 5 runs of 30 s were carried out with changing the region of the sample measured for each run.

**Acknowledgments.** The authors would like to thank Håkan Wennerström for helpful discussions.

**Supplementary Materials.** To view supplementary material for this article, please visit http://doi.org/10.1017/qrd.2022.9.

**Financial support.** This work was supported by the Knut and Alice Wallenberg Foundation (grant number 2016.0074 to SL and ES) and by the Swedish Research Council (grant number 2019-02397 to SL and ES).

**Conflict of interests.** Authors declare no conflict of interests.

**Authors contributions.** K.M., S.L. and E.S. designed the study. K.M. carried out all of the experimental work apart from the FCCS and iFCCS measurements, which were carried out by S.W. K.M. wrote the manuscript with the input from co-authors.

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
