## [Reviewer Report]

*Comments to Author*: The article is undoubtedly of interest, both from the point of view of developing a set of methods for studying the shape and sizes of lipid nanosystems, as well as the study of the influence of a α-synuclein protein on the shape of vesicles.

In the work it is developed two important aspects

(1) The techniques for studying vesicular nanosystems based on the use of a series of complementary methods of optical microscopy/spectroscopy and cryoelectron microscopy it was developed

(2) Interesting and impotant results were obtained on the effect of vesicules shape modulation by the protein containing the amphipathic helices (AHs). This study was performwd on the α-synuclein protein. This protein belongs to the group of AH-containing proteins that induce the division of liposomal nanoparticles. Using the method of cross-correlation spectroscopy (FCCS), the authors showed that in this case this protein does not induce the process of vesicle division. This is perhaps the most important conclusion of this work.

To determine the change in vesicle volume upon addition of α-synuclein, the authors successfully used inverse fluorescence cross-correlation spectroscopy (iFCCS).

The possibilities of cryo-TEM are effectively used. However, the conclusions drawn from this part of the work are not new: in particular, in the article by N. Mizuno et al. (N. Mizunu, et al., 2012, doi.org/10.1074/jbc.M112.365817) was shown that in the presence of α-synuclein, spherical vesicles are converted into cylindrical micelles ∼50 Å in diameter.

The data on the geometrical parameters of vesicles obtained by cryo-electron microscopy are shown in Figs. 4, the meaning of which is not entirely clear. What do the blue boxes in Fig. 4A?, what are the red dots on the histogram? More explanation and description needed.

Currently, drug delivery methods using liposomal formulations are being actively developed. These are liposomes, extracellular vesicles, exosomes, micelles. In the majority, these are complex structures with membranes of different lipid composition, which include proteins, peptides and other biological molecules. All this affects the shape of lipid membranes and determines the shape of lipid nanostructures as a whole. At the same time, questions related to the influence of the shape of such nanosystems on their behavior in living objects remain largely open and unclear. Therefore, the development of methods for controlling the shape of lipid nanosystems and methods for controlling the shape are important.

In the present work, the authors correctly used optical methods and cryoelectron microscopy techniques to characterize vesicles. In particular, the method of cross-correlation spectroscopy (FCCS) was correctly used to prove the integrity of vesicles when the α-synuclein protein was introduced into the membrane.

However, it is somewhat disappointing that, having used the correlation approach, having obtained correlation curves from which it is possible to estimate the size of vesicles, the authors did not do this. This would be interesting, since having a model object with a size and shape determined by another method, it would be possible to evaluate the correctness of using the dynamic light scattering method when measuring the sizes of nonspherical vesicles and obtain quantitative data on determining the geometric parameters of such nanosystems.

Among the methods for measuring the characteristics of vesicles listed by the authors, the method of trajectories is mentioned. Unfortunately, the data obtained by this method are presented in the article rather sparingly and insufficiently compared with other results.

Despite a well-chosen set of experimental methods, their correct use, and important results obtained, the article leaves the impression of some incompleteness. In this work, I would like to see an idea of the mechanism of lipid membrane curvature, supported by a simple computer model. This would provide insight into the molecular mechanisms of the change in membrane curvature upon incorporation of the α-Synuclein protein into the lipid bilayer. It would give an understanding of what is the conformation of α-synuclein in the lipid membrane of this composition. In addition, it would help to substantiate the mysterious conclusion declared in the conclusion of the article: “We put forward a hypothesis that α-synuclein self-assembles on a membrane and forms a supramolecular structure…” It should be noted that the association of the α-synuclein protein in the lipid membrane is highly debatable.

In general, the article is useful and worthy of publication after removing the above remarks.

---

## [Reviewer Report]

*Comments to Author*: This manuscript reports on the investigation of alpha-synuclein-induced small unilamellar vesicle (SUV) deformation aiming to clarify whether this represents fusion intermediate, fission intermediate, or vesicle clustering? Cryo-TEM imaging reveal spherical SUV deformation to prolate spheroids upon the alpha-synuclein binding. Further investigations using fluorescence cross-correlation spectroscopy and nanoparticle tracking analysis showed no signs of fusion, occasional signs of fission. Taken together the results favour a hypothesis that the deformed SUVs are fission intermediates.

This manuscript addresses an important problem related to synaptic plasticity. The results of this work are reasonably comprehensive and conclusions convincing. The manuscript is well-written in terms of structure and style. It can be recommended for publication, provided the authors address my critical comments arranged in order of priority.

1.The inference that the increased diffusion time of the vesicles with alpha-synuclein was due to the structural deformation and increased mass was questionable. The increased diffusion time can be attributed to the decreased volume (70% of the original volume) as measured by the inverse fluorescence cross-correlation spectroscopy. This follows from the Stokes-Einstein equation: the diffusion coefficient is inversely proportional to a particle diameter.

2.What is the rationale for choosing fluorescence cross-correlation spectroscopy to distinguish vesicles labelled with red or green fluorescence dye?If discrete fluorescent vesicles were detectable, the authors would be able to observe the colour mixing by using fluorescence microscopy in case of vesicle fusion. In general, it is advisable to introduce FCCS and iFCCS in one paragraph to place these techniques in the context of the existing analytical techniques.

3.Fig. 5A. 2D cross-section areas of the SUVs in the presence and absence of alpha-synuclein were equal. Is it consistent with the iFCCS results showing the vesicle volume decreased 70% of the original volume?

4.Fig. 1. No distorted spheres are observable in 100% DOPS in 10 mM MES, similar to the pure water case.Is it consistent with the explanation of the hypertonic osmotic imbalance during the sample preparation?

5.Has the increase of the outer layer curvature due the alpha-synuclein been established or just hypothesised by the authors? It is not clearly spelled out.

6.Why do the authors infer that SUVs deform to prolate spheroid, not oblate spheroid?

7.DOPC:DOPS - provide full name.

8.I suggest recolouring the blue curves and circles to green to match to the green fluorescence. It is more intuitive.

---

## [Reviewer Report]

*Comments to Author*: Reviewer #1: This manuscript reports on the investigation of alpha-synuclein-induced small unilamellar vesicle (SUV) deformation aiming to clarify whether this represents fusion intermediate, fission intermediate, or vesicle clustering? Cryo-TEM imaging reveal spherical SUV deformation to prolate spheroids upon the alpha-synuclein binding. Further investigations using fluorescence cross-correlation spectroscopy and nanoparticle tracking analysis showed no signs of fusion, occasional signs of fission. Taken together the results favour a hypothesis that the deformed SUVs are fission intermediates.

This manuscript addresses an important problem related to synaptic plasticity. The results of this work are reasonably comprehensive and conclusions convincing. The manuscript is well-written in terms of structure and style. It can be recommended for publication, provided the authors address my critical comments arranged in order of priority.

1.The inference that the increased diffusion time of the vesicles with alpha-synuclein was due to the structural deformation and increased mass was questionable. The increased diffusion time can be attributed to the decreased volume (70% of the original volume) as measured by the inverse fluorescence cross-correlation spectroscopy. This follows from the Stokes-Einstein equation: the diffusion coefficient is inversely proportional to a particle diameter.

2.What is the rationale for choosing fluorescence cross-correlation spectroscopy to distinguish vesicles labelled with red or green fluorescence dye?If discrete fluorescent vesicles were detectable, the authors would be able to observe the colour mixing by using fluorescence microscopy in case of vesicle fusion. In general, it is advisable to introduce FCCS and iFCCS in one paragraph to place these techniques in the context of the existing analytical techniques.

3.Fig. 5A. 2D cross-section areas of the SUVs in the presence and absence of alpha-synuclein were equal. Is it consistent with the iFCCS results showing the vesicle volume decreased 70% of the original volume?

4.Fig. 1. No distorted spheres are observable in 100% DOPS in 10 mM MES, similar to the pure water case.Is it consistent with the explanation of the hypertonic osmotic imbalance during the sample preparation?

5.Has the increase of the outer layer curvature due the alpha-synuclein been established or just hypothesised by the authors? It is not clearly spelled out.

6.Why do the authors infer that SUVs deform to prolate spheroid, not oblate spheroid?

7.DOPC:DOPS - provide full name.

8.I suggest recolouring the blue curves and circles to green to match to the green fluorescence. It is more intuitive.

Reviewer #2: The article is undoubtedly of interest, both from the point of view of developing a set of methods for studying the shape and sizes of lipid nanosystems, as well as the study of the influence of a α-synuclein protein on the shape of vesicles.

In the work it is developed two important aspects

(1) The techniques for studying vesicular nanosystems based on the use of a series of complementary methods of optical microscopy/spectroscopy and cryoelectron microscopy it was developed

(2) Interesting and impotant results were obtained on the effect of vesicules shape modulation by the protein containing the amphipathic helices (AHs). This study was performwd on the α-synuclein protein. This protein belongs to the group of AH-containing proteins that induce the division of liposomal nanoparticles. Using the method of cross-correlation spectroscopy (FCCS), the authors showed that in this case this protein does not induce the process of vesicle division. This is perhaps the most important conclusion of this work.

To determine the change in vesicle volume upon addition of α-synuclein, the authors successfully used inverse fluorescence cross-correlation spectroscopy (iFCCS).

The possibilities of cryo-TEM are effectively used. However, the conclusions drawn from this part of the work are not new: in particular, in the article by N. Mizuno et al. (N. Mizunu, et al., 2012, doi.org/10.1074/jbc.M112.365817) was shown that in the presence of α-synuclein, spherical vesicles are converted into cylindrical micelles ∼50 Å in diameter.

The data on the geometrical parameters of vesicles obtained by cryo-electron microscopy are shown in Figs. 4, the meaning of which is not entirely clear. What do the blue boxes in Fig. 4A?, what are the red dots on the histogram? More explanation and description needed.

Currently, drug delivery methods using liposomal formulations are being actively developed. These are liposomes, extracellular vesicles, exosomes, micelles. In the majority, these are complex structures with membranes of different lipid composition, which include proteins, peptides and other biological molecules. All this affects the shape of lipid membranes and determines the shape of lipid nanostructures as a whole. At the same time, questions related to the influence of the shape of such nanosystems on their behavior in living objects remain largely open and unclear. Therefore, the development of methods for controlling the shape of lipid nanosystems and methods for controlling the shape are important.

In the present work, the authors correctly used optical methods and cryoelectron microscopy techniques to characterize vesicles. In particular, the method of cross-correlation spectroscopy (FCCS) was correctly used to prove the integrity of vesicles when the α-synuclein protein was introduced into the membrane.

However, it is somewhat disappointing that, having used the correlation approach, having obtained correlation curves from which it is possible to estimate the size of vesicles, the authors did not do this. This would be interesting, since having a model object with a size and shape determined by another method, it would be possible to evaluate the correctness of using the dynamic light scattering method when measuring the sizes of nonspherical vesicles and obtain quantitative data on determining the geometric parameters of such nanosystems.

Among the methods for measuring the characteristics of vesicles listed by the authors, the method of trajectories is mentioned. Unfortunately, the data obtained by this method are presented in the article rather sparingly and insufficiently compared with other results.

Despite a well-chosen set of experimental methods, their correct use, and important results obtained, the article leaves the impression of some incompleteness. In this work, I would like to see an idea of the mechanism of lipid membrane curvature, supported by a simple computer model. This would provide insight into the molecular mechanisms of the change in membrane curvature upon incorporation of the α-Synuclein protein into the lipid bilayer. It would give an understanding of what is the conformation of α-synuclein in the lipid membrane of this composition. In addition, it would help to substantiate the mysterious conclusion declared in the conclusion of the article: “We put forward a hypothesis that α-synuclein self-assembles on a membrane and forms a supramolecular structure…” It should be noted that the association of the α-synuclein protein in the lipid membrane is highly debatable.

In general, the article is useful and worthy of publication after removing the above remarks.